# Barriers to uptake and use of pre-exposure prophylaxis (PrEP) among communities most affected by HIV in the UK: findings from a qualitative study in Scotland

Ingrid Young,[1] Paul Flowers,[2] Lisa M McDaid[1]

[1]MRC/CSO Social and Public Health Sciences Unit, University of Glasgow, Glasgow, UK
[2]School of Health and Life Sciences, Glasgow Caledonian University, Glasgow, UK

**Correspondence to**
Dr Ingrid Young;
ingrid.young@glasgow.ac.uk

## ABSTRACT

**Objectives:** To explore the acceptability of pre-exposure prophylaxis (PrEP) among gay, bisexual and men who have sex with men (MSM) and migrant African communities in Scotland, UK.

**Design:** Consecutive mixed qualitative methods consisting of focus groups (FGs) and in-depth interviews (IDIs) explored PrEP acceptability. Data were digitally recorded, transcribed and analysed thematically to identify anticipated and emerging themes.

**Setting:** Participants were recruited through community sexual health and outreach support services, and from non-sexual health settings across Scotland.

**Participants:** Inclusion criteria included identification as either MSM and/or from migrant African communities; 18 years and older; living in Scotland at the time of participation. 7 FGs were conducted (n=33): 5 with MSM (n=22) and 2 mixed-sex groups with African participants (n=11, women=8), aged 18–75 years. 34 IDIs were conducted with MSM (n=20) and African participants (n=14, women=10), aged 19–60 years. The sample included participants who were HIV-positive and HIV-negative or untested (HIV-positive FG participants, n=22; HIV-positive IDI participants, n=17).

**Results:** Understandings of PrEP effectiveness and concerns about maintaining regular adherence were identified as barriers to potential PrEP uptake and use. Low perception of HIV risk due to existing risk management strategies meant few participants saw themselves as PrEP candidates. Participants identified risk of other sexually transmitted infections and pregnancy as a concern which PrEP did not address for either themselves or their sexual partners. PrEP emerged as a contentious issue because of the potentially negative implications it had for HIV prevention. Many participants viewed PrEP as problematic because they perceived that *others* would stop using condoms if PrEP was to become available.

**Conclusions:** PrEP implementation needs to identify appropriate communication methods in the context of diverse HIV literacy; address risk-reduction concerns and; demonstrate how PrEP can be part of a safe and comprehensive risk management strategy.

### Strengths and limitations of this study

- Our study offers new insights into the psychological and social barriers to pre-exposure prophylaxis (PrEP) uptake and use.
- This paper identifies how limited understandings of PrEP effectiveness and wider community attitudes affect PrEP acceptability.
- We demonstrate the need to consider socially embedded sexual risk management strategies, which address concerns beyond the risk of HIV transmission.
- We suggest there is a need to consider how clinical research is translated into real world contexts and communicated to potential users, as well as how potential users are supported in integrating this information into existing risk management strategies.

Pre-exposure prophylaxis (PrEP) represents a significant biomedical addition to international HIV prevention efforts. The clinical efficacy of the use of antiretrovirals (ARVs) by HIV-negative individuals to prevent the sexual transmission of HIV has been extensively reported.[1 2] The acceptability of PrEP to potential PrEP users (or candidates) to date has focused largely, although not exclusively, on the risk of inefficient PrEP use, such as low or inconsistent adherence to the medication and reduced condom use. It has also focused predominately on experiences in the USA.[3–5] There are currently a number of PrEP demonstration studies underway in the UK, Australia, the USA and elsewhere to explore how it might be used in 'real world' settings.[6 7] Despite being available in the USA since the Food and Drug Administration (FDA) approved truvada (emtricitabine/tenofovir disoproxil fumarate) for use as PrEP in 2012, there has been

relatively low uptake among targeted communities affected by HIV, including gay, bisexual and other men who have sex with men (MSM).[6] Moreover, PrEP has emerged as a controversial issue among those affected by HIV[8 9] and there are clear global inequalities in terms of access to PrEP as it is still only available off-label in a number of countries.

There is a need to understand how uptake and use of PrEP as an HIV prevention strategy could be affected by a range of factors, such as community attitudes towards PrEP and the role of PrEP within concurrent HIV prevention strategies of potential PrEP users.[10 11] We report on the first qualitative study in the UK of the acceptability of PrEP among populations most affected by HIV to inform targeted PrEP implementation strategies.

## METHODS

This article describes a mixed qualitative methods study on the acceptability of PrEP and treatment as prevention (TasP) with two communities in Scotland: (1) MSM and (2) men and women from migrant African communities (first generation immigrants including asylum seekers, students and other migrants). HIV is largely concentrated in these two communities in the UK, as they represent the highest levels of HIV prevalence.[12 13] This article reports on findings in relation to barriers to PrEP use. Our sample included HIV-negative and/or untested participants as potential PrEP users and HIV-positive participants as the potential or existing sexual partners of PrEP users. We include findings from HIV-positive participants because of evidence which indicates that serodiscordant sexual partnerships may be an important factor in PrEP acceptability.[14] HIV-positive individuals may shape PrEP uptake through influence on community attitudes, as well as more directly on potential HIV-negative sexual partners.

First, exploratory focus groups (FGs) were conducted with a convenience sample of MSM and African participants between August and November 2012 to identify community attitudes and emerging issues around PrEP acceptability. We conducted seven FGs: five with MSM (n=22), and two mixed sex groups with African participants (n=11). Three FGs were conducted with HIV-positive MSM (n=14) and one FG with HIV-positive Africans (n=8). Participant age ranged between 18 and 75 years and discussions took place in urban and semiurban locations across central Scotland. Participants were recruited through existing community and/or support groups with the assistance of sexual health and/or LGBT (lesbian, gay, bisexual and transgender) organisations. FG discussion topics included existing risk management strategies in sexual health and an exploration of PrEP and TasP (see box 1).

We then conducted 34 in-depth interviews (IDIs) between March and September 2013 with a purposive sample of MSM (n=20) and African participants (n=14) to explore issues emerging from FG findings and examine personal risk management practices in further depth. Half of the IDI participants were HIV negative or untested for HIV, and the other half had been diagnosed with HIV at the time of the interview (MSM, n=10; Africans, n=7). Inclusion criteria did not specify risk behaviour to allow for broad exploration of candidacy factors, including those not related to sexual behaviour. IDI participants were aged 19–60 years and were resident in four Scottish regions (Glasgow, Lothian, Lanarkshire and Grampian). Tailored flyers and posters were distributed to: workshops and support groups; community sexual health testing clinics; gay bars, saunas and clubs; an MSM mail-out condom programme; commercial venues (eg, African food and barber shops) and in educational settings known to have large African student populations. Community organisations also made contact with potential participants through their regular online and face-to-face outreach and support work. Interviews took place in private spaces, in partner organisations or in participants' own homes, and focused the acceptability of PrEP and TasP, including awareness, potential use, concerns and combination with other and/or existing risk management strategies (such as TasP, serosorting, etc) (see box 2).

PrEP was explained to participants by drawing on but not limited to the use of a visual aid (figure 1). Basic explanations of PrEP were consistent across all FG and IDI discussions (box 2, section 4a). Subsequent and more detailed descriptions of PrEP varied depending on participant questions, which were encouraged and answered. This approach was taken to identify how PrEP should be described to potential candidates. Material did not specify an exact efficacy rate due to the emerging clinical data, variability according to adherence,

---

> **Box 1    Focus group discussion guide**
>
> *Part 1*
> Participants were presented with a number of objects to discuss. Objects included: condoms, sachets of lubricant, pregnancy test, list of antiretrovirals, mocked up bottle of antibiotics, empty boxes/bottles of truvada, and pictures of: an Oraquick® In-Home HIV Test and rapid HIV tests.
> - What do these objects make you think about?
> - How do these objects relate to risk and HIV?
> - What is risky in relation to HIV?
> - Do you use these objects to manage HIV?
> - What else do you use to manage HIV?
>
> *Part 2*
> Provide visual cards of pre-exposure prophylaxis and treatment as prevention provided and explain separately.
> - How might you use these pills?
> - How do you think your friends or sexual partners might use these pills?
> - Are you concerned about the use of pills as a form of HIV prevention?
> - Would these pills change the way people currently manage HIV? What do you think about this?

**Box 2    Interview topic guide for HIV-negative or untested participants**

1. Experiences with and/or proximity to HIV
   ► Is HIV a risk for you?
   ► Is HIV something that you talk about with your sexual partners? (or friends?)
   ► Have you ever tested for HIV?
2. Risk management/sexual health
   ► What are the risks for you in relation to sexual health?
   ► You've said that you manage risk in sexual health by…Has this changed and how?
   ► Do you talk to your sexual partners about how you manage your sexual health?
   ► Do you use health or other services to help you manage your sexual health? If so, how?
3. Use of existing technologies
   ► *List of sexual health technologies physically presented to participant*:
      − Condoms
      − HIV testing
      − Sexually transmitted infections (STIs) testing
      − Contraception (the pill, intrauterine devices, long-term injections, etc)
      − Pregnancy testing
      − CD4 counts
      − Viral loads
      − Antiretrovirals (ARVs)
      − Postexposure prophylaxis (PEP)
   ► Do you use any of these now? Have you used any of these in the past?
   ► How have you used them? What made you use them?
   ► Have you used any of these in combination with other prevention methods?
   ► How do you feel about using them?
   ► If you started/stopped using some of these, can you say why you did?
4. Potential use of new technologies
A. Pre-exposure prophylaxis (PrEP)
   ► *Approximate PrEP description explained to participant:*
      PrEP is when ARVs are used by people who are HIV negative to prevent the transmission of HIV. At the moment, it can be taken once a day, although researchers are looking into other forms (short-term PrEP, long-acting injectable, topical gel/microbicides). PrEP only works if people take the medication regularly. Clinical trials have shown that it is effective in relation to how often people take the pills. People are still encouraged to use condoms and other forms of risk reduction with PrEP use. There are some side effects, but this may not affect everyone and it generally seems to be well tolerated in the clinical trials. PrEP is not currently available in the UK, but it has been licenced for use in the USA.
   ► Have you heard of PrEP before?
   ► What do you think of PrEP as a prevention method?
   ► How would you feel about using PrEP as a prevention method?
   ► If you would use PrEP, how do you think you would you use it?
   ► Do you have any concerns about PrEP as a prevention method?
   ► Do you think other people might use PrEP as a method?
B. Treatment as prevention (TasP)
   ► *Approximate TasP description explained to participant:*
      TasP is when ARVs are used by people living with HIV not only to clinically manage HIV, but also to help prevent the transmission of HIV. TasP manages the 'viral load' or the amount of HIV in the system. Research has shown that having an 'undetectable' viral load means that transmission of HIV is unlikely to happen. So, if the HIV positive person is taking their treatment regularly, and they do not have an STI, and their viral load is 'undetectable' for a period of time (about 6 months), they would clinically be considered not infectious. TasP in particular is when treatment is started to prevent transmission, rather than when the person clinically needs the treatment. If someone starts this treatment early—including for prevention reasons—they cannot stop taking this medication. So although people living with HIV will eventually move onto treatment, perhaps after 5 to even 10 years, this would mean starting treatment considerably earlier.
   ► Have you heard of TasP before?
   ► What do you think of TasP as a prevention method?
   ► Can you imagine using this as a prevention method with a sexual partner who is HIV positive?
   ► How would you feel if a sexual partner suggested this as an HIV prevention method?
   ► Do you have any concerns about this as a prevention method?
   ► How do you think other people who are HIV negative or untested might feel about using ARVs or HIV treatment as a prevention method?

## PrEP (Pre-Exposure Prophylaxis)

| | |
|---|---|
| **Who is it for?** | Person <u>not living with HIV</u> (HIV negative) |
| **What is it?** | ARVs are taken **before exposure to HIV** (usually before having sex with someone living with HIV) |
| **What does this involve?** | PrEP would mostly likely be a pill that would be taken everyday. Side effects could include nausea, diarrhoea, headaches, and tiredness. |
| **Does this work?** | PrEP is not 100% effective, but **when used with condoms, significantly reduces risk of HIV transmission**. |
| **Can I get it here?** | No, PrEP is not currently available in Scotland or anywhere else in the UK. |

**Figure 1** Pre-exposure prophylaxis (PrEP) Visual Aid. ARVs, antiretrovirals.

and to not overly complicate the explanation. Participants were informed that PrEP efficacy was dependent on levels of adherence, as demonstrated in a number of trials. FG participants were told that the iPrEx study reported approximately 73% protection if taken regularly (90% adherence), which was accurate at the time discussions were conducted.[15] IDI participants were informed that efficacy could be up to or more than 90% if taken regularly, drawing on subsequent subanalyses of clinical findings.[2] Participants were informed that other forms of risk reduction were recommended, such as condoms.[16] Efficacy of condoms was described as less than 100%.[17 18] Discussions explored a wide-range of PrEP scenarios, including non-condom use.

Written consent was provided by all participants at the start of the FGs and IDIs. All FGs and IDIs were digitally recorded and transcribed verbatim. Transcripts were anonymised and coded in NVivo V.10. Data were analysed thematically, drawing on anticipated as well as emergent themes.[19–21] Rigour throughout the analysis was achieved through an iterative process of discussion and revision between coauthors.[19 20 22]

## RESULTS

We identified five potential barriers to effective PrEP use: interpreting effectiveness; managing adherence; PrEP candidacy and low perceptions of HIV risk; concerns with other risks such as the criminalisation of HIV transmission and sexually transmitted infections (STIs); and moral barriers. We have identified extracts taken from FGs; otherwise, it can be assumed that the extract comes from an IDI participant.

### Interpreting effectiveness

Understandings of PrEP effectiveness emerged as an important barrier to potential and effective use. Although participants were informed that PrEP was highly effective when taken regularly, most participants expressed concerns that it provided less than 100% protection and therefore was 'insufficient' to prevent HIV transmission on its own. Some participants felt that PrEP used in isolation was 'too much' of a risk (if only approximately 70% effective).

*Respondent 1*: That's not enough, that's not enough, exactly.

*Respondent 2*: Its seventy-two per cent effective but then there's still that twenty-eight per cent. (HIV-negative MSM, FG)

Participants also expressed a wider scepticism of PrEP, which emerged in two ways. Some thought that the advice to use condoms with PrEP indicated a continued scientific uncertainty: "I don't know if this [will] work because since they said if you use it you can still use condom, that means they are not sure as well" (HIV-negative African woman). Second, some expressed scepticism in relation to the variability in reported efficacy rates in relation to adherence. As a result, there was a reluctance to trust these, unless the information came from a recognised, trusted and reliable source.

*R1*: There's so much out there right, on the internet an' everything else right, if it came up on the news right? The logistic news on BBC one that this pill will prevent so…—like a hundred per cent—that's when I would believe it would work. I [would not] trust any other pill…

*Question*: So it has to be a hundred per cent?

*R2*: 'cause there's a lot a…no like—anyone can put something on the internet, …You've got to use like certain websites, you've got to look like .org, .gov, those type o' websites. You've just done one that's like www.medicine. co.uk it's not a…it's not like a legitimate website it could just be anyone postin' stuff up there.

*Q*: Well I suppose in the States it's from the Food and Drug Administration so it's got that stamp on it.

*R1*: Yes, but you [don't know] if they're been a hundred per cent accurate on their job or just been lazy an' the fact that we're brave enough to take it to human trials right? (HIV-negative MSM, FG)

In spite of being told that the FDA had approved the drug, these FG participants did not recognise the organisation or trust it as a reliable source, and were therefore sceptical about its statements and/or endorsement.

Other participants were less concerned about PrEP not being 100% efficacious because they imagined that it would be used *in addition* to condoms. However, participants expressed confusion or uncertainty in knowing how to interpret efficacy rates and their impact on risk reduction in this particular context. For example, in discussing the effectiveness of condoms in relation to other prevention strategies, one participant calculated the protection offered by PrEP, condoms and TasP with a hypothetical HIV-positive sexual partner. Assuming that PrEP was 90% effective at reducing HIV transmission, he explained:

> R: but there's obviously still a 10% risk but, as you said, there's the same risk with condoms. So it's either you take the 10% risk or you say 'well, I'll use condoms and we'll use TASP' which makes 180%. (Laugh)
>
> Q: And PrEP. (Laugh)
>
> R: 253 out of 300! (HIV-negative MSM)

Although the participant in this extract was joking about adding up the numbers, the interpretation of efficacy rates in relation to other prevention options posed a potential source of confusion or misinformation for participants.

### Managing adherence

Given the evidence concerning the patterning of efficacy by adherence,[2] maintaining regular adherence to medication was identified as a potential barrier to effective PrEP use. Some participants described how they might forget to take tablets or their routine might be disrupted because of non-regular working patterns, and were therefore concerned about how effective PrEP would be if they did not take the drugs regularly: "Sometimes I forget. Because I don't have a regular sleeping pattern, so sometimes I'll fall asleep at four in the afternoon and I usually take my pills [for an existing health condition] after dinner" (HIV-negative MSM). Establishing and maintaining a routine to take PrEP was also affected by the perception of social stigma attached to HIV medication. This meant maintaining daily adherence would be difficult if there was limited privacy (eg, the presence of roommates) or if there was a change in environment. One participant reported concerns about his family potentially finding the tablets:

> If I was on those medications for a year and if I went home…or anything like that I'd find it very difficult to be able to take my medications or I would find it a bit of a barrier that if my family knew about it they'd investigate why are you on these pills. And again that would probably put some doubt in their head and then they'd probably then think the worst—that I was HIV positive. (HIV-negative MSM)

In addition to being unable to establish or maintain a regular routine to facilitate good adherence, most participants expressed concerns about the physical effects of the drugs themselves, and how the side-effects might inhibit taking them. Some participants described disliking taking tablets and that this would be a barrier to daily PrEP use: "I don't really like pills so I don't think I would take any pills every day" (HIV-negative African man). Many viewed the potential for side-effects, such as nausea and diarrhoea, as too great a trade-off for increased HIV prevention, in spite of being informed that trials had reported high tolerance of the drugs:

> But, could I put up wi' side-effects all the time as well, like taking a pill, and going tae a nightclub, and ended up wi' diarrhoea? You know, you've gotta think of things like that, am I [going to] take this? Am I [going to] stand tired, have a headache in a club just at the fact that I might at the end of it get a shag…(HIV-negative MSM)

Other participants felt the side-effects would interfere with sexual practice itself: "Nausea, an' diarrhoea an'—you'll be shittin' all over his [penis]. No [thanks]." (HIV-negative MSM, FG). A large minority of HIV-positive as well as HIV-negative participants raised the issue of longer term side-effects, especially those who were more familiar with the effects of ARVs:

> Does it not have any side effects inside my body? 'Cause from my reading as well, some of these anti-retrovirals have got side-effects with the lungs, with the kidneys and stuff. So, in the long run, if I get used to drinking this pill and I'm actually not exposed because I'm thinking I might be at risk of getting exposed, am I not doing more damage to myself in my body? (HIV-negative African woman)

### PrEP candidacy and low perceptions of HIV risk

Many participants described scepticism in taking PrEP daily, especially if they were not always exposed to or at risk of HIV transmission. As such, perceptions of HIV transmission risk played an important role in potential uptake of PrEP. Several participants rejected the use of PrEP because they perceived themselves to be at a very low risk. For the majority of HIV-negative participants, this was because most adopted serosorting or believed they could accurately tell if a potential partner was HIV positive and avoid sex with him or her: "I wouldn't just [want to] take a risk because it's already defining as a risk, I mean, it's a risk, it's already been defined as a risk and since I'm a little bit risk averse…I won't just do anything" (HIV-negative African man). HIV stigma and assumptions about disclosure appeared to inform these strategies. For example, some HIV-negative participants assumed that HIV-disclosure to sexual partners "was the law" (HIV-negative MSM, FG), and that their sexual partners were not HIV positive. Those HIV-negative participants in serodiscordant relationships and HIV-positive participants suggested that condoms, and the nature of their relationships (eg, monogamous), already helped manage their risks, and that the additional benefit from

PrEP was unnecessary. One man in a serodiscordant relationship explained:

> Right now in my current situation as, I'm doing a monogamous relationship and everything like that, again I don't think even in that [instance] I don't think I would take the pills even though it would be an extra measure…I think I would feel comfortable enough in the current situation. (HIV-negative MSM)

### PrEP and concerns with other risks

Both HIV-positive and HIV-negative participants identified risk of other STIs as a concern for themselves and their sexual partners. Heterosexual African participants, especially African women, also talked of the risk of pregnancy. For those participants who were either living with HIV, or who had experience of serodiscordant relationships, other health risks posed to the HIV-positive sexual partner through PrEP use were also identified:

> when I'm looking at condom use it's not just HIV that you're protecting yourself against but some other STIs as well. So you're killing two birds with one stone, more like it. But when it's now using this, you're actually getting yourself involved in something that you already know —'this person is positive and I'm [going to] have sex with him, but it's okay, I will not use condoms because I know I can use this.' But you're also putting that person, because that person's immune system might be weak. They're also putting that person at risk of other STIs as well, instead of using condoms. So I think it makes people more ignorant of the other things as well. And it makes people just more focused on just HIV and not other STIs. And then risk of pregnancy as well is likely to get high, provided the people are not on any other form of contraception….No, I think…no, from the way you've said it to me…I wouldn't even tell it to people. (HIV-negative African woman)

Most HIV-positive participants described anxieties about PrEP in relation to *their* inability to be in effective control of HIV prevention. One woman could not imagine agreeing to her HIV-negative sexual partner using PrEP: "[PrEP] is too risky for him because I don't know when he stop using it, what will happen to him" (HIV-positive African woman, FG).

### Moral barriers to PrEP

PrEP emerged as a highly contentious issue because of the perceived negative implications it had for existing risk management strategies and HIV prevention. Many participants viewed PrEP as problematic because they perceived that *others* would stop using condoms if PrEP was available. That is, they distinguished between how *they* might use PrEP and how they perceived others would use it. For some, the concern was in relation to reduced condom use: "it'll be like…women burning their bras. It'll be all these guys [whipping] off their condoms, do you know what I mean?" (HIV-negative MSM). Other participants were also concerned with

bigger changes in sexual practice, such as increased risk taking:

> Are you trying to eliminate the condom?…There is no place for a condom at all when it's like this. So I think it actually encourages people to be more promisc…not promiscuous, but to be more…I don't know [if] 'ignorant' is the right word, but to be less careful 'cause they know 'oh, there's that pill'. And it's not even cost-effective than just getting a condom or using…PEP once in a blue moon. (HIV-negative African woman)

Not all participants were opposed to PrEP and many viewed PrEP as a good addition to HIV prevention options. However, there was still concern about *how* PrEP would be accessed. Overall it was hoped that it would be made available only under strict conditions to ensure the 'right' people received it and that it was used in the 'correct' way:

> I believe it should be available in Scotland. But, it should be under strict control to make sure the person who takes it knows that he, or she is not protected to full extent from other infections, and HIV either. (HIV-negative MSM)

## DISCUSSION

This is the first qualitative study in the UK to report on the acceptability of PrEP. Our research offers new insights into the psychological and social barriers to PrEP uptake and use. We identified the significance of how the effectiveness of PrEP as a risk-reduction intervention is communicated to and understood by potential candidates. Our research suggests the importance of HIV risk perception and found that, for many participants, PrEP was not immediately seen as a trusted and/or beneficial addition to their repertoire of existing risk-reduction practices. Our findings also highlight how existing risk management strategies in relation to PrEP encompass broad concerns relating to sexual health, relationships, social factors and communities.

Understanding how to interpret PrEP efficacy rates, on their own and in combination with other prevention strategies, proved a stumbling block for the participants and poses a considerable challenge to how health providers support the concept of combination prevention in the context of PrEP. Liu *et al*[23] identify accurate consumer knowledge as key to PrEP implementation, in addition to addressing other factors such as stigma, adherence and risk reduction. While we agree with Liu, our findings suggest that the form and delivery of this consumer knowledge, including *how* health providers understand and communicate this information, needs further attention to support effective PrEP use. Communicating PrEP effectiveness in real world settings will be a two-way process that demands clarity on the part of providers and potential users. In addition to supporting providers, negotiating PrEP as a prevention strategy will require improved levels of HIV literacy among

potential PrEP users to be effective. We suggest the need to consider *critical* HIV literacy, which encompasses the ability to know, understand and use HIV-related information within existing risk-reduction practices.[24] As levels of HIV knowledge are directly affected by a range of factors, such as proximity to HIV,[25] inequalities in HIV literacy within communities affected by HIV will play an important role in understanding the barriers to PrEP use. These factors have direct implications for both the nature of how PrEP-related HIV risk prevention is delivered and by whom.

For many participants, PrEP was not seen as a necessary or welcome addition to their repertoire of risk management strategies. Our findings suggest that risk perception and candidacy will play a critical role in decisions to use PrEP, a finding echoed by Golub *et al*.[26] For some of the HIV-negative and/or untested participants in our study, the rejection of PrEP as unnecessary emerged from a perception that they were not at risk of HIV transmission. HIV risk was managed often through the sexual exclusion of HIV-positive sexual partners or through reliance on monogamous sexual relationships. These findings suggest that PrEP implementation strategies will need to engage with these wider, socially embedded risk-reduction practices, including how HIV stigma might affect risk perception. Moreover, our research highlights how the management of risk was not limited only to HIV transmission, but also to the risk of STIs, pregnancy and social stigma. Participants described how PrEP would not adequately address these existing risks, and even had the potential to create significant new risks. Saberi *et al*[27] reported similar moral concerns about the implications of PrEP on condom use in their study with MSM participants in serodiscordant relationships in the USA. Although they surmise that proximity to HIV and age play a role in these concerns, our findings suggest that moral objections to PrEP were not limited by age or sexuality. We suggest that these moral reactions to PrEP as a risk-reduction option are related to broader social and community concerns about the potential for PrEP to radically change the way prevention is practiced.[9] This highlights the need to engage with wider social concerns about what constitutes 'inappropriate' or high-risk sexual practice in relation to PrEP and to demonstrate how PrEP implementation can be a part of a safe and comprehensive risk management strategy.

Our study has a number of strengths and limitations. We employed rigorous qualitative methodology which enabled in-depth exploration of the social meanings of PrEP acceptability and likely use.[21] We therefore add to the existing quantitative PrEP acceptability research. With a small sample of MSM and migrant African participants in a non-generalised HIV epidemic, some of whom were engaged in sexual health or community services, our findings are not generalisable to a wider population but we would argue, are transferable to similar populations in similar social contexts. As we did not sample according to sexual risk behaviour, our findings are only transferrable to broad risk groups and not necessarily to 'high risk' individuals. Our inclusion of the recommendation to use condoms in the visual PrEP information may have biased the findings. However, our discussions included consistent and broad descriptions of PrEP and encompassed a wide range of PrEP scenarios, including non-condom use.

Our research identifies the need to consider acceptability factors which extend beyond drug adherence and risk compensation when introducing and scaling up PrEP and has a number of implications for policy and clinical practice. In particular, it will be necessary to develop clear tools and techniques to communicate PrEP information to potential candidates, and to support health providers in the implementation of these tools. These methods will need to translate clinical research relating to PrEP effectiveness in real world contexts as is appropriate in the context of diverse critical literacy skills. Implementation will also need to address low-risk perception, non-HIV-related risk reduction and other moral concerns to demonstrate how PrEP can be a part of a safe and comprehensive risk management strategy. Our study suggests that broader social and community concerns about the potential for PrEP to change HIV prevention need to be addressed by supporting the integration of PrEP into existing risk management strategies and through targeted promotion.

**Acknowledgements** The authors thank the organisations (Waverly Care, Terrence Higgins Trust Scotland, LGBT Youth Scotland, Gay Men's Health Scotland, Positive Scotland, NHS Greater Glasgow and Clyde, NHS Lothian, NHS Grampian) who helped with recruitment and the men and women who agreed to take part in the focus groups and interviews.

**Contributions** IY designed the study, carried out the qualitative data collection and analysis and drafted the manuscript. PF helped to design the study, carried out some of the data collection and analysis and helped to draft the manuscript. LMD participated in the design of the study and helped to draft the manuscript. All authors read and approved the final manuscript.

**Funding** The HIV and the Biomedical Study, IY and LMD are core funded by the UK Medical Research Council (MRC) (MC_U130031238/MC_UU_12017/2) at the MRC/Chief Scientist Office (CSO) Social and Public Health Sciences Unit, University of Glasgow. PF is funded by Glasgow Caledonian University.

**Competing interests** None.

**Patient consent** Obtained.

**Ethics approval** College of Social Sciences Ethics Committee, University of Glasgow (CSS2012/0193, CSS20120264).

**Provenance and peer review** Not commissioned; externally peer reviewed.

**Data sharing statement** This study was undertaken by the MRC/CSO Social and Public Health Sciences Unit, University of Glasgow. Our data sharing policies comply with that of the UK Medical Research Council. Unpublished data from this study is available on request and information about the study and data is made available on the MRC/CSO Social and Public Health Sciences website. https://www.sphsu.mrc.ac.uk/research-programmes/sh/shfsub/accbiomhivprev.html.

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
