## [Reviewer comments · BMJ Open]

Some articles will have been accepted based in part or entirely on reviews undertaken for other BMJ Group journals. These will be reproduced where possible.

ARTICLE DETAILS

TITLE (PROVISIONAL)	Barriers to uptake and use of Pre-Exposure Prophylaxis (PrEP) amongst communities most affected by HIV in the UK: Findings from a qualitative study in Scotland
AUTHORS	Young, Ingrid; Flowers, Paul; McDaid, Lisa

VERSION 1 - REVIEW

REVIEWER	Martin Holt UNSW Australia, Australia
REVIEW RETURNED	06-Jun-2014

GENERAL COMMENTS	This is a clearly written paper that will be a useful addition to the field. To strengthen the paper, I recommend the following: Participants. I would like the authors to provide a bit more detail about the participants, reflecting on whether they were likely candidates for PrEP or not e.g. provide a description of their relationships statuses, sexual practices etc. The participants' membership of MSM or African communities is taken as evidence that they might be potential users of PrEP, but as many of the participants appear to have suggested, they are not at sufficient risk of HIV to want or need PrEP (this aligns quite nicely with most of the quantitative acceptability research about who is interested in PrEP). Did the authors consider targeting recruitment at men and women at high risk of HIV, referring to their sexual practices, for example? If not, I think this is a limitation, as the participants are really expressing general views about PrEP (i.e. community attitudes to PrEP), not the views of people who would seriously consider taking it. Future research might be worth conducting with those who appear to be in most need. Efficacy of different techniques. I think you should explain how you explained the efficacy of existing techniques (e.g. condoms) in relation to PrEP to participants, as this would heavily influence their assessment of whether PrEP would be useful or risky. Reviews of condoms in the 'real world' suggest about 80% effectiveness, but most people think it is higher. If condoms are positioned as most effective, PrEP is likely to be seen as suspect, particularly by people who are not particularly at risk of/close to HIV or who don't have problems with condoms. It's interesting to see that some participants picked up that Figure 1 implies that PrEP is less effective than condoms, because it is recommended that PrEP be used with condoms. I suspect this recommendation would be seen as very conservative now. I suggest you comment on how the study's explanation of the strategies may have influenced the participants' views.
---

	Moral barriers. This is a really interesting section. The perception that 'others' might ruin the use of PrEP (by taking more risks) or that only the 'right' people should get PrEP is worthy of more attention. We see similar tropes in Australian research about PrEP (and TasP). It really seems to trouble people that others might be able to have sex without condoms and be protected from HIV. Why is that? Maybe you could relate this in the Discussion to the 'PrEP wars' that have been raging in the US and UK (to a lesser extent) about whether PrEP is a useful addition or the end of prevention as we know it? Positive aspects of PrEP. The results seem very focused on the negative aspects of PrEP and barriers to its use. Is this because that's all the participants talked about, or have you quarantined the results about the positive aspects of PrEP for publication elsewhere? Did you put positive aspects of use to participants? If positive views of PrEP weren't expressed, could this be related to the participants not having much need for PrEP? 'HIV literacy' is used in a number of places but not defined until late in the Discussion. I suggest including a brief definition in the abstract and the full definition early in the paper. HIV literacy would also appear to be strongly related to one's level of experience with sex, HIV and risk (p.14), hence my interest in whether the participants engage in practices that would justify PrEP. p.15. When you mention 'inappropriate' sexual practice, I think you need to be explicit that what counts as appropriate conduct may be highly variable. Define TasP at first use.
--	--

REVIEWER	Darrell H. S. Tan St Michael's Hospital & University of Toronto, Canada
REVIEW RETURNED	23-Jul-2014

GENERAL COMMENTS	This qualitative study was a mixed-methods investigation of the opinions of MSM and men/women from migrant African communities in Scotland on the acceptability of PrEP. Overall this straightforward study was well conducted and the manuscript is well and concisely written. The findings raise issues that are helpful and relevant to the potential implementation of PrEP in industrialized world settings. A few issues warrant further consideration, as listed below. MAJOR COMMENTS 1. The methods section states that PrEP efficacy was only discussed if/when participants asked. Given that efficacy is a pivotal piece of information regarding PrEP it is unclear why this was not addressed more directly. Did some focus groups/participants not ask, and hence never receive this information? In addition, the quoted PrEP efficacy of 73% from iPrEx was specifically in the subgroup of individuals with 90% adherence or higher – was this nuance clearly stated to participants? In addition, at the time that the study was conducted, findings from clinical trials among heterosexual African populations were also available (Partners, TDF2, VOICE, FEM-PrEP), and might reasonably be expected to be meaningful to the migrant African populations – was there a reason that these trials were not discussed? These issues are particularly
---

important because effectiveness was highlighted as one of the five key findings of the study.

2. While it is appropriate that HIV-infected participants were included in the study, they represented the majority of participants (2/3 of focus group participants and half of the interviews). Given that PrEP is specifically for use by HIV-uninfected persons please justify this disproportionate representation of HIV-infected perspectives.

3. The term “HIV literacy” is used to describe the first identified barrier in the Results section, which has to do with perceptions of risk, understandings of prevention efficacy, and the trustworthiness of information sources. This term is defined by the authors, but not until the Discussion section. Is there a citation on which this definition is based? Its focus on the use of information within the context of risk reduction practices does not intuitively follow from its name (“HIV literacy” might more intuitively be construed as encompassing understanding germ theory, basics of what HIV does to the body, etc. rather than focusing on prevention). Consider both moving the definition of this term into the results section, and/or revising the term used.

4. In the Discussion section the authors suggest that social inequalities directly affect HIV literacy but it is unclear on what basis this conclusion is based. The corresponding section of the Results does not discuss these social inequalities, and the authors note that they did not sample or analyse the data by education or socioeconomic status.

5. Not perceiving oneself as a PrEP candidate (eg. due to being in a monogamous serodiscordant relationship) and low perceptions of HIV risk are identified as the third potential “barrier” to PrEP. However, these issues should arguably only be labeled as a barrier (which implies that they need to be overcome) if these perceptions are in fact erroneous in some way. For instance, not perceiving oneself to be a PrEP candidate despite being in a monogamous serodiscordant relationship may be quite reasonable if the positive partner is on suppressive antiretroviral therapy and/or if condoms are being used, particularly in light of recent data on the risk of horizontal transmission with undetectable viral load (eg. the PARTNER study), but may more appropriately be construed as a “barrier” to PrEP if the partner is not on ART. Similarly, it might be quite appropriate that an HIV-negative African woman who could not imagine knowingly having an HIV-positive sexual partner not be considered a PrEP candidate, if indeed her practice is to verify HIV negative serostatus in monogamous male partners. Therefore, to what extent was the reality of participants’ situations explored with them during the study, to assess the extent to which they should or should not be realistically considered appropriate for initiation of PrEP, and the extent to which this truly is a “barrier”?

MINOR COMMENTS

1. The first paragraph of the introduction states that PrEP is “not available” in the UK; while it may not be approved by regulatory authorities, presumably it is nevertheless available for off-label usage. Please clarify.

2. The first sentence of the Methods refers to “acceptability of TasP” rather than PrEP; later, on page 6 (line 32), TasP is again listed as a focus of the interviews. As such the research question or study objective is not clearly defined. Please clarify whether the focus of the study was on PrEP, TasP or both.

3. Please clarify further exactly how the inclusion criterion of being “from migrant African communities” was defined and implemented. Were these all first-generation immigrants or could some be children

	of immigrants? Were refugees included in addition to immigrants? Is information available on how long ago they immigrated to the UK? 4. The study employed a mixed-methods approach including focus groups and interviews, which is very reasonable. However the methods section would be strengthened by an explicit statement justifying the use of the two methodologies. 5. Some of the quotations include spellings or non-standard word usage that may represent local dialect, or manners of speech (or potentially typographical errors) but may be unfamiliar to some readers – eg. on page 10, “going tae a nightclub”, “No ta”. Consider using explanatory comments or standardizing spellings as appropriate. 6. The text refers to Scotland as a “low-prevalence setting” which is factually accurate when considering adult HIV seroprevalence in the UK as a whole, but is arguably less accurate when specifically considering this figure for the MSM population to whom PrEP is primarily being targeted. As such it may be more appropriate to refer to the country as an “industrialized world” setting or some other term that implies the concentration of the epidemic in MSM and other specific groups.
--	---

REVIEWER	Rivet Amico University of Michigan
REVIEW RETURNED	26-Jul-2014

GENERAL COMMENTS	Barriers to uptake and use of Pre-Exposure Prophylaxis (PrEP) amongst communities most affected by HIV in the UK. While the methods are generally sound and results supported with appropriate quotes, this reviewer has considerable concerns over if and how the information provided (Figure 1) may have caused the kind of confusion and “low literacy” the authors identify as characterizing PrEP acceptability and factors influencing uptake. Of greatest concern is the information provided to participants to explain what PrEP is that said it was “not 100% effective, but when used with condoms, significantly reduces risk of HIV transmission”. By most accounts and interpretation of the evidence base, even the emerging evidence base available during the conduct of the study, participants may have been misinformed and rightly confused by the information provided. The efficacy of PrEP has been established in trials that also offered a host of HIV-prevention tools and strategies, including HIV testing and provision of condoms. Typically, efficacy of PrEP is described as reduction in HIV-risk when PrEP is offered in the context of comprehensive sexual health protection services, including access to free condoms. Providing condoms does not mean people use them or use them correctly and results of these trials were not restricted to those reporting condom use. To say that PrEP is effective only in the context of using condoms is simply untrue and likely caused individuals to wonder why one would adopt PrEP if he is already using condoms. And, if not using condoms, then they would not consider themselves a contender for PrEP because it would offer no benefit. This is confusing as well as inaccurate. Where the authors position results as potentially indicating poor HIV literacy, they may simply be recalling the
--

information they were provided. Rather than these results presenting opportunities to learn about perceptions and acceptability of PrEP, they seem more interesting from the perspective of what people do with mixed messages about PrEP and condom use. Given the message that PrEP and condoms must be used, how open are people to using PrEP? Do results suggest that we should talk frankly with people that don't use condoms about added protection? Re-analysis of the data within the framework of the information provided (eg., how do people feel about PrEP when this is what they are told or know about it) should be considered. Results, in any case, should be interpreted with these limitations in mind.

Specific comments to consider are included below.

Abstract: "gay, bisexual and men who have sex with men (MSM) and migrant..." is confusing. How is 'gay' different from MSM? Inclusion criteria noted below this suggests MSM or migrant.

Abstract: Typo- missing period after 'emerging themes' in design.

Abstract: As noted above, HIV literacy is not clearly the issue.

Introduction: Note that many of the demonstration projects are more reflective of how individuals may use an open-label product versus use outside of a research study.

Introduction: The introduction feels light or cursory in terms of reflecting current evidence base and research on perceptions of acceptability of PrEP in communities. Further the argument for the added value of looking at this in a low-prevalence setting needs to be fleshed out more clearly.

Methods: page 5 notes acceptability of TasP but PrEP, or treatment for prevention, seems to be more appropriate here.

Methods: page 5, line 16- partners would read better as "partnerships"

Methods: The authors need to repeat and expand on inclusion criteria- presently it only appears in the abstract and does not indicate how "African migrant" was operationalized for inclusion or whether sexually active was considered.

Methods: Please note if groups were single gender or mixed.

Methods: Include main questions from the interviews in a table.

Methods: As noted above, the information sheet has questionable information and likely influenced responses considerably. This should be presented as such. Also, in light of this, when this information was provided in the interview is important.

Methods: Not sure where the 73% protection is coming from in the Baeten and Celum manuscript cited. The as treated value in their table 2 is 92% based on drug level.

	Methods: Page 6. Not clear what “(as before)” is referring to, Results: Page 7- Define “R” and “Q” for readers once. Results: Page 8- Framing participants “imagined” that PrEP would be used in addition to condoms seems inappropriate. The participants were told that, based on the information figure. Thus, it would not be so much that it was imagined but rather they were reiterating what they had been instructed. Note quote on page 12 seems to suggest this “..from the way you’ve said it to me...” Results: Not sure that ‘moral’ is the right term here. Could the authors provide some reasoning towards why this is ‘moral’ versus fears of increased risk for STIs or concerns about potential reduction in condom use or behavioral disinhibition or risk compensation. Discussion: Authors note work was with potential PrEP users and their HIV-positive sexual partners but there was no presentation of work specific to couples. Discussion: The authors note HIV literacy as an issue but it is not clear from the results that literacy about HIV is at issue versus being well informed about PrEP. Discussion: Typo- extra “.” after communities on line 23 page 14. Discussion: The authors note that more than consumer information is needed for informed decisions- which certainly would not be argued against by anyone. They do note that Liu et al consider information key and position that work as being inconsistent with that observation. Liu et al note several additional factors- including adherence, risk reduction, stigma and access. The author’s presentation of Liu’s work seems inaccurate. Discussion: The authors conclude that social inequalities impact health literacy. A number of references would support this assumption. However, their statements in the discussion lack detailed exploration of what they mean or how they would envision this being an issue and suggest resolution. They argue that information alone is insufficient and also argue the education/information is critical. Better presentation in this section to provide guidance that synthesizes findings is needed. Discussion: The authors note VOICE study results “attest to” risk perception being critical in successful use of PrEP. There are a number of problems with this statement. VOICE results did not clearly establish a role of feeling at risk and study-product use. Rather, product use was low and people generally felt at low risk for HIV in that cohort. More importantly, adherence was to a blinded, placebo controlled study product and not PrEP. Non-use of study product appears to have had a lot to do with mistrust in the community and efforts to secure services available to study participants that were otherwise unavailable in the community while concomitantly not wanting to take a tablet or use gel. The role of
--	---

	perceived risk or “need” for PrEP is critical but the use of VOICE results to buttress the argument is misplaced. Discussion: Line 39 page 15- what do the authors mean by “inappropriate or high-risk sexual practice”? Discussion: The final paragraph does not seem to offer novel content or compelling conclusions. While the authors note the need to consider factors beyond adherence and risk compensation, it would be helpful to list out what factors were identified in the research in summary and how that guides the future research noted as presently needed. Figure 1: As noted in general comments, this figure suggests a fair amount of mixed information. Individuals were informed of PrEP as a hypothetical (most likely be a pill [note typo is this line]) and further would likely involve a number of side effects. There is no indication that these would be limited to a start-up syndrome or notes that it is actually fairly uncommon. The wording for efficacy is extremely confusing. When used with condoms PrEP would be largely unnecessary except for the case of misuse of condoms or breakage. Guidelines adopt wording that says that it was effective in the context of providing a number of risk reduction support tools and strategies, including free condoms. PrEP trial participants were not required to take them or use them or use them correctly. To anchor efficacy of PrEP to its protective effects when also using condoms is incorrect and likely caused people to wonder why one would need PrEP if they are consistent condom users. This reviewer has serious concerns over the use of this information sheet and how it may have biased findings and thus limit generalizability of results.
--	--

VERSION 1 – AUTHOR RESPONSE

Reviewer Name Martin Holt

Participants. I would like the authors to provide a bit more detail about the participants, reflecting on whether they were likely candidates for PrEP or not e.g. provide a description of their relationships statuses, sexual practices etc. The participants' membership of MSM or African communities is taken as evidence that they might be potential users of PrEP, but as many of the participants appear to have suggested, they are not at sufficient risk of HIV to want or need PrEP (this aligns quite nicely with most of the quantitative acceptability research about who is interested in PrEP). Did the authors consider targeting recruitment at men and women at high risk of HIV, referring to their sexual practices, for example? If not, I think this is a limitation, as the participants are really expressing general views about PrEP (i.e. community attitudes to PrEP), not the views of people who would seriously consider taking it. Future research might be worth conducting with those who appear to be in most need.

- Participants were recruited from the community with no explicit inclusion criteria on sexual risk behaviour. This was a broad, exploratory study and sought to include community attitudes to PrEP, as well as specific acceptability issues from potential candidates. We felt that acceptability issues are broader than risk behaviour and sought to identify what these were. We have in fact shown that social

factors in risk management play an important role in identifying as potential PrEP candidates. Moreover, some of the candidates did consider themselves or their (potential) sexual partners as PrEP candidates. While we have added this as a potential limitation to the study, we have also made it more explicit that this research was seeking to understand how community attitudes to PrEP play an important role in acceptability, at both the individual and community level, and that factors other than risk behaviour will affect perceptions of candidacy. See pages 4 – 5.

Efficacy of different techniques. I think you should explain how you explained the efficacy of existing techniques (e.g. condoms) in relation to PrEP to participants, as this would heavily influence their assessment of whether PrEP would be useful or risky. Reviews of condoms in the 'real world' suggest about 80% effectiveness, but most people think it is higher. If condoms are positioned as most effective, PrEP is likely to be seen as suspect, particularly by people who are not particularly at risk of/close to HIV or who don't have problems with condoms. It's interesting to see that some participants picked up that Figure 1 implies that PrEP is less effective than condoms, because it is recommended that PrEP be used with condoms. I suspect this recommendation would be seen as very conservative now. I suggest you comment on how the study's explanation of the strategies may have influenced the participants' views.

- Figure 1 was used as an aid to explain PrEP and was not the only source of information. In all the focus groups and interviews, participants were presented with the figure, and PrEP was further explained verbally. We have added a section to the methodology on how PrEP was explained and included our interview topic guide which further elaborates how we addressed PrEP (See page 6 – 7). PrEP was explained as an option to reduce risk of HIV transmission, based on how much it was taken. Participants were informed that studies found that if the drugs were taken regularly (between 4 – 7 times a week), HIV transmission was found to be reduced by more than 90% (based on the subanalyses from iPrEX, amongst others). We were also clear with participants that condoms were not 100 % effective (even when used correctly) (see p7) and included this in the new section. The decision to say that condoms were still recommended was based on CDC guidance from 2011 which said: 'Provide risk-reduction and PrEP medication adherence counseling and condoms. This does not say condoms have to be used, but that condoms should be provided along with other risk reducing advice. However, much of the literature on the acceptability of PrEP focuses on risk compensation, which explicitly refers the abandonment of condoms. These issues were discussed in focus groups and interviews as participants responded to the explanation of PrEP. This element of the study highlights the challenges to translating clinical trial findings into real world messages that make sense to potential PrEP users (and their potential HIV-positive sexual partners), which is a key finding in our research.

Moral barriers. This is a really interesting section. The perception that 'others' might ruin the use of PrEP (by taking more risks) or that only the 'right' people should get PrEP is worthy of more attention. We see similar tropes in Australian research about PrEP (and TasP). It really seems to trouble people that others might be able to have sex without condoms and be protected from HIV. Why is that? Maybe you could relate this in the Discussion to the 'PrEP wars' that have been raging in the US and UK (to a lesser extent) about whether PrEP is a useful addition or the end of prevention as we know it?

- We were pleased this was of interest to the reviewer. We have added in a reference to the PrEP wars and commented on how reactions to PrEP as a risk reduction option are related to broader debates and concerns about HIV prevention and the potential for PrEP to radically change the way

prevention is supported and practiced. See p 16.

Positive aspects of PrEP. The results seem very focused on the negative aspects of PrEP and barriers to its use. Is this because that's all the participants talked about, or have you quarantined the results about the positive aspects of PrEP for publication elsewhere? Did you put positive aspects of use to participants? If positive views of PrEP weren't expressed, could this be related to the participants not having much need for PrEP?

- The focus of paper is explicitly on barriers, given size of paper and emerging findings from this study. We do have some data on facilitators of PrEP which will be explored elsewhere. However, one of the main findings was the how participants problematised PrEP and initially rejected it as a viable prevention option. We felt that the majority of issues to emerge were potential barriers and we suggest these barriers need to be addressed. In essence, we felt our study identified factors which will help to better target and support PrEP users. We have explicitly stated that this article, based on a larger study, reports on barriers. See page 5.

'HIV literacy' is used in a number of places but not defined until late in the Discussion. I suggest including a brief definition in the abstract and the full definition early in the paper. HIV literacy would also appear to be strongly related to one's level of experience with sex, HIV and risk (p.14), hence my interest in whether the participants engage in practices that would justify PrEP.

- This relates to wider issues about how PrEP was understood in the study. We recognize that we have not defined HIV literacy. However, upon reflection, we feel it is more accurate to explore issues in relation to interpreting efficacy in the results, and to then draw out the implications of this in relation to HIV literacy in the discussion. We have therefore changed the sub-title of this section in the results and modified the abstract accordingly. We have also been clearer about our definition of HIV literacy. In particular, we hoped to show how inequalities in HIV literacy was not simply a lack of information (or provision of misinformation) but more about how participants used/applied the information to their own personal risk management practices, in combination with other risk reduction strategies, and that this will be a critical issue when considering how best to support the introduction of PrEP. See p 15.

p.15. When you mention 'inappropriate' sexual practice, I think you need to be explicit that what counts as appropriate conduct may be highly variable.

- We used inappropriate as defined by participants. We acknowledge the ambiguity of using this, and have qualified the wording accordingly. See page 16.

Define TasP at first use.

- We have done this. See page 5.

Reviewer Name Darrell H. S. Tan

MAJOR COMMENTS

1. The methods section states that PrEP efficacy was only discussed if/when participants asked. Given that efficacy is a pivotal piece of information regarding PrEP it is unclear why this was not addressed more directly. Did some focus groups/participants not ask, and hence never receive this information? In addition, the quoted PrEP efficacy of 73% from iPrEx was specifically in the subgroup

of individuals with 90% adherence or higher – was this nuance clearly stated to participants? In addition, at the time that the study was conducted, findings from clinical trials among heterosexual African populations were also available (Partners, TDF2, VOICE, FEM-PrEP), and might reasonably be expected to be meaningful to the migrant African populations – was there a reason that these trials were not discussed? These issues are particularly important because effectiveness was highlighted as one of the five key findings of the study.

- As we explained above, when we provided statistics on efficacy rates to participants, we explained that these were based on levels of adherence. We drew not only on iPrEX but also on other PrEP trials in communicating clinical findings to participants. However, we did not explicitly name the trials, given the complexity of the trials and their findings, the time available to the interview, but also wanted to spend more time talking about the way in which this risk reduction intervention might be used in real world settings. Participants exhibited little interest in the details of the trials. Given the nature of this qualitative research (in-depth, semi-structured interviews), we engaged in lengthy and individualised discussions with participants to explain how PrEP worked. While we provided efficacy rates, we spoke about the relationship between how often pills were taken and how effective they were at potentially reducing risk of transmission. We felt this method was appropriate for this study, which was exploring the broader role of biomedical HIV prevention techniques. As outlined above, we have further explained how we explained PrEP to participants in the methods section. See page 6 – 7.

2. While it is appropriate that HIV-infected participants were included in the study, they represented the majority of participants (2/3 of focus group participants and half of the interviews). Given that PrEP is specifically for use by HIV-uninfected persons please justify this disproportionate representation of HIV-infected perspectives.

- Firstly, this was not a stand-alone PrEP study, but was also exploring issues relating to TasP. Secondly, this article reports on two phases of this study – exploratory focus groups and in-depth interviews. We acknowledge that the majority of FG participants were living with HIV and that this disproportionately represents views on PrEP for this analysis. However, the FG phase of the study was exploratory and identified critical themes which we further explored in the interview phase. In line with qualitative methodology and our approach to this phase, we did not think it necessary to have absolutely even numbers of HIV-negative/untested participants, as we were also exploring a number of other variables which might affect acceptability, such as geography, age, etc. In the interview phase, we included an equal number of participants who were HIV negative, providing a more representative sample in terms of analysis.

- The inclusion of people living with HIV in this study is based on the rationale that we were exploring the wider attitudes of people from communities affected by HIV, and including the potential (or existing) sexual partners of PrEP users. As we have outlined, we felt it was of critical importance to explore how both sexual parties responded to this HIV prevention method, as PrEP will not be used in a vacuum. As our findings have shown, PrEP is an intervention that would be discussed, in some cases, with HIV-positive sexual partners. For many of the HIV positive participants, PrEP was not an intervention which they trusted, and therefore, poses a significant barrier to those serodiscordant couples who might otherwise be considered eligible for PrEP. We therefore feel that our inclusion of HIV positive participants more broadly in the study is justified. To this extent, we have included a further explanation of the inclusion of these participants. See page 5.

3. The term “HIV literacy” is used to describe the first identified barrier in the Results section, which has to do with perceptions of risk, understandings of prevention efficacy, and the trustworthiness of information sources. This term is defined by the authors, but not until the Discussion section. Is there a citation on which this definition is based? Its focus on the use of information within the context of risk reduction practices does not intuitively follow from its name (“HIV literacy” might more intuitively

be construed as encompassing understanding germ theory, basics of what HIV does to the body, etc. rather than focusing on prevention). Consider both moving the definition of this term into the results section, and/or revising the term used.

- While we feel the findings raise issues in relation to HIV literacy, we agree with the reviewer that the term can be misleading as we originally described it. As explained above, we have removed it from the title of the results section, and have added to the discussion about the implication of interpreting effectiveness on wider HIV literacy issues. See page 15.

4. In the Discussion section the authors suggest that social inequalities directly affect HIV literacy but it is unclear on what basis this conclusion is based. The corresponding section of the Results does not discuss these social inequalities, and the authors note that they did not sample or analyse the data by education or socioeconomic status.

- The social inequalities we identified were in relation to serostatus, geography, sexuality, migration status, etc. We have modified the discussion to more clearly explain which inequalities we are referring to and how these inequalities affect HIV literacy and response to PrEP. See page 15.

5. Not perceiving oneself as a PrEP candidate (eg. due to being in a monogamous serodiscordant relationship) and low perceptions of HIV risk are identified as the third potential “barrier” to PrEP. However, these issues should arguably only be labeled as a barrier (which implies that they need to be overcome) if these perceptions are in fact erroneous in some way. For instance, not perceiving oneself to be a PrEP candidate despite being in a monogamous serodiscordant relationship may be quite reasonable if the positive partner is on suppressive antiretroviral therapy and/or if condoms are being used, particularly in light of recent data on the risk of horizontal transmission with undetectable viral load (eg. the PARTNER study), but may more appropriately be construed as a “barrier” to PrEP if the partner is not on ART. Similarly, it might be quite appropriate that an HIV-negative African woman who could not imagine knowingly having an HIV-positive sexual partner not be considered a PrEP candidate, if indeed her practice is to verify HIV negative serostatus in monogamous male partners. Therefore, to what extent was the reality of participants’ situations explored with them during the study, to assess the extent to which they should or should not be realistically considered appropriate for initiation of PrEP, and the extent to which this truly is a “barrier”?

- We entirely agree that perception of low risk, as described by some participants, was entirely accurate and therefore, their rejection of PrEP was appropriate. However, the interviews explored in depth the nature of sexual relationships and how risk was deduced. When a participant said they eliminated all HIV positive sexual partners, they were asked how they did that. For example, did they discuss status with sexual partners? Did they share test results, etc. Given that our research questions addressed both PrEP and TasP acceptability, we explored a range of scenarios where the HIV-positive partner may well have an undetectable viral load and probed in relation to PrEP use. However, in addition to the possibility of actual low risk due to other risk management strategies, and therefore, potentially limited suitability for PrEP, a number of participants described their sexual practice as low risk based on the tacit assumption that their sexual partners were not living with HIV. We interpret these findings in relation to HIV stigma, and assumptions about the sexual practice of people living with HIV, and the assumption that everyone will know their status. We feel this highlights wider social disparities between HIV-negative and HIV-positive experiences in relation to risk perception. We have further explained our methods to account for these issues, including our focus group and interview topic guides. (See Tables 1 and 2).

MINOR COMMENTS

1. The first paragraph of the introduction states that PrEP is “not available” in the UK; while it may not be approved by regulatory authorities, presumably it is nevertheless available for off-label usage. Please clarify.

- We have clarified that PrEP is only available off-label in the UK. As such, there is currently no indication about its use apart from access via trials. See page 4.

2. The first sentence of the Methods refers to “acceptability of TasP” rather than PrEP; later, on page 6 (line 32), TasP is again listed as a focus of the interviews. As such the research question or study objective is not clearly defined. Please clarify whether the focus of the study was on PrEP, TasP or both.

- As we have explained above, this study explored both PrEP and TasP acceptability and that this article focuses specifically on barriers to PrEP. See page 5.

3. Please clarify further exactly how the inclusion criterion of being “from migrant African communities” was defined and implemented. Were these all first-generation immigrants or could some be children of immigrants? Were refugees included in addition to immigrants? Is information available on how long ago they immigrated to the UK?

- We have included a clearer definition of migrant African communities. See page 5. Given the heterogeneity of the sample (in relation to immigration status and length of time in the country), we do not have information on how long ago each of the participants migrated. This is a hard to reach group in Scotland, with whom very little research has been conducted. Moreover, this was an exploratory study to understand both individual and community acceptability and to identify potential barriers to implementation. We therefore did not feel that a strict sampling frame in terms of time in the country would necessarily access the critical issues in relation to PrEP acceptability.

4. The study employed a mixed-methods approach including focus groups and interviews, which is very reasonable. However the methods section would be strengthened by an explicit statement justifying the use of the two methodologies.

- We have included further description about the use of each phase of the study. We drew on focus groups as a way of identifying emerging and critical issues in relation to PrEP, as well as community attitudes. We then pursued these issues in more in-depth and personal interviews, where we could further explore PrEP acceptability with participants. See pages 5 – 6.

5. Some of the quotations include spellings or non-standard word usage that may represent local dialect, or manners of speech (or potentially typographical errors) but may be unfamiliar to some readers – eg. on page 10, “going tae a nightclub”, “No ta”. Consider using explanatory comments or standardizing spellings as appropriate.

- While we feel the local dialect is important to represent the diversity of the Scottish sample, we recognize the potential miscommunication this may cause. We have thus changed the extracts to standard spelling/wording to ensure clarity of meaning.

6. The text refers to Scotland as a “low-prevalence setting” which is factually accurate when considering adult HIV seroprevalence in the UK as a whole, but is arguably less accurate when specifically considering this figure for the MSM population to whom PrEP is primarily being targeted. As such it may be more appropriate to refer to the country as an “industrialized world” setting or some other term that implies the concentration of the epidemic in MSM and other specific groups.

- We have changed our description to explain that these two groups have the highest rates of prevalence in the UK. See page 5.

Reviewer Name R Amico

Barriers to uptake and use of Pre-Exposure Prophylaxis (PrEP) amongst communities most affected by HIV in the UK. While the methods are generally sound and results supported with appropriate quotes, this reviewer has considerable concerns over if and how the information provided (Figure 1) may have caused the kind of confusion and “low literacy” the authors identify as characterizing PrEP acceptability and factors influencing uptake. Of greatest concern is the information provided to participants to explain what PrEP is that said it was “not 100% effective, but when used with condoms, significantly reduces risk of HIV transmission”. By most accounts and interpretation of the evidence base, even the emerging evidence base available during the conduct of the study, participants may have been misinformed and rightly confused by the information provided. The efficacy of PrEP has been established in trials that also offered a host of HIV-prevention tools and strategies, including HIV testing and provision of condoms. Typically, efficacy of PrEP is described as reduction in HIV-risk when PrEP is offered in the context of comprehensive sexual health protection services, including access to free condoms. Providing condoms does not mean people use them or use them correctly and results of these trials were not restricted to those reporting condom use. To say that PrEP is effective only in the context of using condoms is simply untrue and likely caused individuals to wonder why one would adopt PrEP if he is already using condoms. And, if not using condoms, then they would not consider themselves a contender for PrEP because it would offer no benefit. This is confusing as well as inaccurate.

- We did not inform participants that it was only effective with condoms, but suggested that condoms were still recommended as it was not 100 percent effective. We had discussions at length with participants about how PrEP might be used with other risk reduction strategies (including but not limited to condoms, - e.g. testing, partner selection, specific sexual practices, etc). Most participants appeared to understand that PrEP was only partially effective. While we report on the barriers to PrEP in this paper, a small minority of participants felt that PrEP was an excellent idea and were excited about the potential reduced risk it could provide. As such, we have added a new section which outlines how we explained PrEP, the extent to which we used the visual aid (Figure 1) and how PrEP was describe in the interviews (Table 2). We have also elaborated on how discussions of PrEP were in-depth, and nuanced, and moved beyond using PrEP with or without condoms, but included a range of combination factors. See the methodology section.

Where the authors position results as potentially indicating poor HIV literacy, they may simply be recalling the information they were provided.

- We acknowledge that poor HIV literacy is an inaccurate way of describing the reaction to PrEP in this context. We hoped that our use of HIV literacy would be understood in a wider context, and relate to the ways in which participants took up and applied this information in the context of existing risk management practices. That is, we define HIV literacy as a critical form of literacy in which key HIV concepts and facts are drawn on to establish an informed response to risk of transmission. We have changed the title and description of the first results section and better elaborated on what the implications of the findings are in relation to HIV literacy in the discussion. See page 15 – 16.

Rather than these results presenting opportunities to learn about perceptions and acceptability of

PrEP, they seem more interesting from the perspective of what people do with mixed messages about PrEP and condom use. Given the message that PrEP and condoms must be used, how open are people to using PrEP? Do results suggest that we should talk frankly with people that don't use condoms about added protection? Re-analysis of the data within the framework of the information provided (eg., how do people feel about PrEP when this is what they are told or know about it) should be considered. Results, in any case, should be interpreted with these limitations in mind.

- We agree and have modified how these results have been framed. We see the findings from these results as contributing exactly to this issue – to exploring how open people are to using PrEP, to exploring how frank we should be about non-condom use and other possibilities. In our research, although we presented the possibility of condoms to improve risk reduction, we explored the use of PrEP without condoms with most participants. Some participants – as seen in the moral barriers section – however, were adamant that PrEP was not acceptable because it would do away with condoms. We feel that this research highlights how contentious these issues are at a community level, as well as within communities of health practitioners who informed the shaping of this study. We therefore have modified/made clearer how we introduced: the questions of this research, how PrEP was explained, and how we can interpret the results.

Specific comments to consider are included below.

Abstract: “gay, bisexual and men who have sex with men (MSM) and migrant...” is confusing. How is ‘gay’ different from MSM? Inclusion criteria noted below this suggests MSM or migrant.

- We use this phrasing to avoid the conflation of mode of transmission with identity. As we sought out community attitudes, we felt it was important not to reduce participants to an MSM category when they actively identify as gay or bisexual (which was very important for many of our participants). We draw here on work by Young and Meyer (2005) who are broadly critical of the category of MSM as it is currently used. Thus, we define our use of MSM as broadly inclusive of both identity categories, and sexual practice.

Abstract: Typo- missing period after ‘emerging themes’ in design.

- We have changed this.

Abstract: As noted above, HIV literacy is not clearly the issue.

- We have addressed the issues of HIV literacy above, and have removed this term from the abstract.

Introduction: Note that many of the demonstration projects are more reflective of how individuals may use an open-label product versus use outside of a research study.

- We acknowledge this and have rewritten this element of the introduction to more accurately reflect this fact. See page 4.

Introduction: The introduction feels light or cursory in terms of reflecting current evidence base and research on perceptions of acceptability of PrEP in communities. Further the argument for the added value of looking at this in a low-prevalence setting needs to be fleshed out more clearly.

- We refer to a number of comprehensive literature reviews on PrEP acceptability which broadly summarise the evidence. We have added in further references, but feel that the broad description of

research from current acceptability literature is accurate. Moreover, this research focuses specifically on communities experiences concentrated HIV epidemics, who are currently being considered as potential PrEP candidates.

Methods: page 5 notes acceptability of TasP but PrEP, or treatment for prevention, seems to be more appropriate here.

- This was a typo. Moreover, we have further outlined the nature of the study and clarified focus on PrEP and TasP as part of the broader research. See page 5.

-

Methods: page 5, line 16- partners would read better as “partnerships”

- We have changed this to partnerships.

Methods: The authors need to repeat and expand on inclusion criteria- presently it only appears in the abstract and does not indicate how “African migrant” was operationalized for inclusion or whether sexually active was considered.

- As outlined above, we have further explained our use of African migrant. See page 5.

Methods: Please note if groups were single gender or mixed.

- We have clarified that FGs with African participants were mixed sex. See page 5.

Methods: Include main questions from the interviews in a table.

- We have included both topics guides from the focus groups and the interview with HIV negative participants. Our topic guides for interviews with HIV-positive participants was very similar, although with a slightly different emphasis/order. We felt it was unnecessary duplication to include both guides.

Methods: As noted above, the information sheet has questionable information and likely influenced responses considerably. This should be presented as such. Also, in light of this, when this information was provided in the interview is important.

- See our comments above in relation to how we explained PrEP, why we chose to explain it in this way, and how this explanation was situated in wider discussions about PrEP and risk management strategies. See also our new section on how PrEP was introduced. See page 6 – 7.

Methods: Not sure where the 73% protection is coming from in the Baeten and Celum manuscript cited. The as treated value in their table 2 is 92% based on drug level.

- This came from Grant’s NEJM paper and is the figure we used in Focus groups. We have changed the reference accordingly. See the above section on PrEP explanations on pages 6 -7.

Methods: Page 6. Not clear what “(as before)” is referring to,

- We have removed this and clarified FG and IDI content.

Results: Page 7- Define “R” and “Q” for readers once.

- We have done this.

Results: Page 8- Framing participants “imagined” that PrEP would be used in addition to condoms seems inappropriate. The participants were told that, based on the information figure. Thus, it would not be so much that it was imagined but rather they were reiterating what they had been instructed.

- As outline above, we explicitly explore attitudes to condom use with PrEP and were clear that although PrEP was not 100 % effective, regular adherence meant it was quite high. Our interview discussions were highly nuanced and addressed the complexity of using other risk reduction strategies.

Note quote on page 12 seems to suggest this “..from the way you’ve said it to me...”

- For most participants, we were the first point of information about PrEP and spent a considerable amount of time explaining how it worked. This is valuable information about how participants respond to the descriptions we provided and we have discussed this in the discussion, especially in relation to both critical HIV literacy and moral barriers.

Results: Not sure that ‘moral’ is the right term here. Could the authors provide some reasoning towards why this is ‘moral’ versus fears of increased risk for STIs or concerns about potential reduction in condom use or behavioral disinhibition or risk compensation.

- There was much discussion about the responsibility of communities in relation to HIV prevention, and that non-condom use – and by extension PrEP use - was often seen as irresponsible behaviour. A critical theme to emerge was the belief that condom use was responsible sexual behaviour, in spite of other risk reduction options and many participants were highly judgemental about potential PrEP users. We felt that it was important to distinguish between other risks, and moral concerns emanating from the community.

Discussion: Authors note work was with potential PrEP users and their HIV-positive sexual partners but there was no presentation of work specific to couples.

- We did not work specifically with couples but spoke to participants about their current sexual partners, if they had any. We have modified this line to be clear that this is about serodiscordant partnerships more generally (and not limited to monogamous partnerships SD partnerships). See page 5.

Discussion: The authors note HIV literacy as an issue but it is not clear from the results that literacy about HIV is at issue versus being well informed about PrEP.

- Please see our response above in relation to HIV literacy and explanations of PrEP.

Discussion: Typo- extra “.” after communities on line 23 page 14.

- We have modified this.

Discussion: The authors note that more than consumer information is needed for informed decisions-

which certainly would not be argued against by anyone. They do note that Liu et al consider information key and position that work as being inconsistent with that observation. Liu et al note several additional factors- including adherence, risk reduction, stigma and access. The author's presentation of Liu's work seems inaccurate.

- We acknowledge this comment and have changed the reference to Liu's work accordingly. Our aim was to demonstrate how efforts to support PrEP, in addition to the list above, also needs to explore how this information is provided, and how ongoing support to understand and apply this new information needs to be considered. See page 15 – 16.

Discussion: The authors conclude that social inequalities impact health literacy. A number of references would support this assumption. However, their statements in the discussion lack detailed exploration of what they mean or how they would envision this being an issue and suggest resolution. They argue that information alone is insufficient and also argue the education/information is critical. Better presentation in this section to provide guidance that synthesizes findings is needed.

- We have clarified what we mean by social inequalities and how they are related to HIV literacy. See page 15 – 16.

Discussion: The authors note VOICE study results “attest to” risk perception being critical in successful use of PrEP. There are a number of problems with this statement. VOICE results did not clearly establish a role of feeling at risk and study-product use. Rather, product use was low and people generally felt at low risk for HIV in that cohort. More importantly, adherence was to a blinded, placebo controlled study product and not PrEP. Non-use of study product appears to have had a lot to do with mistrust in the community and efforts to secure services available to study participants that were otherwise unavailable in the community while concomitantly not wanting to take a tablet or use gel. The role of perceived risk or “need” for PrEP is critical but the use of VOICE results to buttress the argument is misplaced.

- We acknowledge this point and have removed reference to VOICE results. We have modified this section to more accurately describe the importance of risk perception and included a more appropriate reference in relation to risk identification and PrEP. See page 16.

Discussion: Line 39 page 15- what do the authors mean by “inappropriate or high-risk sexual practice”?

- We have drawn from participants comments to mean non-condom use, and have clarified this fact. See page 16 – 17.

Discussion: The final paragraph does not seem to offer novel content or compelling conclusions. While the authors note the need to consider factors beyond adherence and risk compensation, it would be helpful to list out what factors were identified in the research in summary and how that guides the future research noted as presently needed.

- We have revised the final paragraph to make recommendations appropriate with our findings.

Figure 1: As noted in general comments, this figure suggests a fair amount of mixed information. Individuals were informed of PrEP as a hypothetical (most likely be a pill [note typo is this line]) and

further would likely involve a number of side effects. There is no indication that these would be limited to a start-up syndrome or notes that it is actually fairly uncommon.

- As outlined above, PrEP was explained to participants when presented with the card – that symptoms were uncommon and were found to be eliminated fairly soon after start. This was explained verbally in focus groups, as well as interviews and is noted in the Interview focus guide (Table 2).

The wording for efficacy is extremely confusing. When used with condoms PrEP would be largely unnecessary except for the case of misuse of condoms or breakage.

- We acknowledge this point and have given our explanation as to why we used these terms and how prep was explained earlier on. We stress that this was an aid and are open to its modification for future use. In addition, we have added this to the limitation section of the paper. See page 17.

Guidelines adopt wording that says that it was effective in the context of providing a number of risk reduction support tools and strategies, including free condoms. PrEP trial participants were not required to take them or use them or use them correctly. To anchor efficacy of PrEP to its protective effects when also using condoms is incorrect and likely caused people to wonder why one would need PrEP if they are consistent condom users. This reviewer has serious concerns over the use of this information sheet and how it may have biased findings and thus limit generalizability of results.

- We feel that our methods (explained) included nuanced discussions with participants about PrEP and its role in risk reduction. While we have explained our methods and attempts to address these broader issues, we have also included this as a potential limitation. See page 17.

VERSION 2 – REVIEW

REVIEWER	Martin Holt Centre for Social Research in Health, UNSW Australia
REVIEW RETURNED	30-Sep-2014

GENERAL COMMENTS	I am satisfied that the authors have addressed my earlier comments and congratulate them on a thorough revision of their paper.
---

REVIEWER	Darrell Tan St Michael's Hospital, Toronto, Canada
REVIEW RETURNED	30-Sep-2014

GENERAL COMMENTS	In general the authors have responded thoughtfully to the reviewers' comments. However a few residual issues warrant further consideration:  1. Page 6: "Inclusion criteria did not specify risk behaviour to allow for findings in relation to candidacy" – unclear what this means. 2. In the Methods section, the description of how PrEP was explained to participants has been modified and described in much greater detail in response to reviewers' comments, but almost to the point of obfuscating what was actually communicated to participants. The section would benefit from a summary statement highlighting the key message about efficacy that was communicated to participants, before providing further details. 3. A related point is that the lengthy description of PrEP in the Methods section, taken together with the findings on the "interpreting
---

effectiveness” section of the Results, highlights another key problem that is not discussed enough in the paper – namely, that a key barrier is likely the difficulty that healthcare providers and/or authorities will have in communicating the concept of combination prevention in a clear and understandable way. This concept is briefly alluded to at the beginning of the Discussion (“We identified the significance of how the effectiveness of PrEP... is communicated”) but it is mostly absent from the following paragraph, which emphasizes that understanding on the part of consumers is the major “stumbling block”. Communication of PrEP efficacy/effectiveness is a two-way process that will require clarity and understanding on the part of both providers and users.

4. A previously raised concern was that not perceiving oneself as a PrEP candidate and low perceptions of HIV risk should not necessarily be labeled as a “barrier” to PrEP, since many people are indeed at low risk of acquiring HIV. The authors have responded to this comment by describing how and why perceptions of HIV risk can be erroneous, assumptive and related to HIV stigma. These points are appropriate and valid, but should be raised directly in the paper; otherwise it still seems incomplete to simply label these issues as “barriers”.

5. In the concluding paragraph, what is “non-HIV risk reduction”?

In general the authors have responded thoughtfully to the reviewers’ comments. However a few residual issues warrant further consideration:

1. Page 6: “Inclusion criteria did not specify risk behaviour to allow for findings in relation to candidacy” – unclear what this means.
2. In the Methods section, the description of how PrEP was explained to participants has been modified and described in much greater detail in response to reviewers’ comments, but almost to the point of obfuscating what was actually communicated to participants. The section would benefit from a summary statement highlighting the key message about efficacy that was communicated to participants, before providing further details.
3. A related point is that the lengthy description of PrEP in the Methods section, taken together with the findings on the “interpreting effectiveness” section of the Results, highlights another key problem that is not discussed enough in the paper – namely, that a key barrier is likely the difficulty that healthcare providers and/or authorities will have in communicating the concept of combination prevention in a clear and understandable way. This concept is briefly alluded to at the beginning of the Discussion (“We identified the significance of how the effectiveness of PrEP... is communicated”) but it is mostly absent from the following paragraph, which emphasizes that understanding on the part of consumers is the major “stumbling block”. Communication of PrEP efficacy/effectiveness is a two-way process that will require clarity and understanding on the part of both providers and users.
4. A previously raised concern was that not perceiving oneself as a PrEP candidate and low perceptions of HIV risk should not necessarily be labeled as a “barrier” to PrEP, since many people are indeed at low risk of acquiring HIV. The authors have responded to this comment by describing how and why perceptions of HIV risk can be erroneous, assumptive and related to HIV stigma. These points are appropriate and valid, but should be raised directly in the paper; otherwise it still seems incomplete to simply label these issues as “barriers”.
5. In the concluding paragraph, what is “non-HIV risk reduction”?

REVIEWER	Rivet Amico University of Michigan USA
REVIEW RETURNED	08-Oct-2014

GENERAL COMMENTS	Barriers to uptake and use of pre-exposure prophylaxis (PrEP) amongst communities most affected by HIV in the UK: Findings from a qualitative study in Scotland. Bmjopen-20114-005717.R1 This revised manuscript addressed many of the concerns from the first review. The structure is much improved and the presentation more comprehensive and clear. Suggestions provided below. Abstract: Consider changing “identified barriers to PrEP uptake and use” to “identified barriers to potential PrEP uptake and use” as the work was focused on intentions and hypothetical use rather than responses to an actual opportunity to use PrEP. Article summary: The authors note that the paper identifies inequalities in HIV literacy but the results of the paper do not suggest inequalities, rather they suggest low levels of accurate PrEP information. Introduction: The authors provide a rationale for inclusion of HIV-positive individuals in the focus groups/IDIs but it seems to fall short of reaching a logical conclusion. While serodiscordant partnerships would be a targeted group for PrEP, why attitudes and beliefs from an individual living with HIV are important should rest with their influence in supporting partners or dissuading them. Consider a slight revision to call that out specifically- eg., opinions of HIV-positive individuals will shape PrEP uptake through influence on the community as well as more directly on potential HIV-negative partners. If phrased something like that, there is no need for the sentence that follows (...We did not explicitly...). Methods: Consider moving “We have identified extracts...” from method to the results section. Results: Page 11- Consider using a different word than “significant” in “A significant minority...” Results: Page 11- Consider moving “Many participants described skepticism...” to the next section “PrEP candidacy” Results: Page 12- The authors describe discourse here and elsewhere (page 13) as “could not imagine” which has a ring of judgment to it and feels a bit extreme and stigmatizing. Rather, it seems that most adopted serosorting or believed they could both accurately tell if a potential partner was HIV positive and avoid sex with him/her. Discussion: In a few places, the authors note that “we demonstrated” but really the study was not designed to demonstrate, rather it was to explore, characterize or identify. Thus, the authors explored, identified, characterized, and results suggest, highlight, identify, so on, but do not conclusively demonstrate. Discussion: The authors note “Our findings suggest that HIV literacy is directly affected by geography...” It is not clear what findings are being used to support this claim. The results do not provide support for this. Discussion- Page 16- typo (line 54) Tables: Adding in items from the guides are very helpful. However, presently that table reads like a protocol, or procedural document that the interviewer/facilitator would have used for implementation. The reader does not need that level of detail. Readers will want to be able to see either a list of questions or a select example of the
---

	kinds of items asked. This should not take up several pages but rather be an actual table with themes and sample items. The authors should consolidate current text to make this a brief set of questions or set of example questions for major areas of inquiry.
--	---

VERSION 2 – AUTHOR RESPONSE

Reviewer 2: Darrell Tan, St Michael's Hospital, Toronto, Canada

In general the authors have responded thoughtfully to the reviewers' comments. However a few residual issues warrant further consideration:

1. Page 6: "Inclusion criteria did not specify risk behaviour to allow for findings in relation to candidacy" – unclear what this means.

- We did not wish to predefine PrEP candidate according to risk behaviour as the aim of this study was to better identify all factors and/or barriers to potential PrEP candidacy. We have changed the wording of this sentence to explain that we did not limit participation by current risk behaviour so as to broadly explore a range of factors (including non sexual behaviour) which might affect candidacy. (See page 6).

2. In the Methods section, the description of how PrEP was explained to participants has been modified and described in much greater detail in response to reviewers' comments, but almost to the point of obfuscating what was actually communicated to participants. The section would benefit from a summary statement highlighting the key message about efficacy that was communicated to participants, before providing further details.

- We have modified the interview topic guide (Table 2) to explicitly include a brief explanation of PrEP which was described to participants. We have referred to this specifically by indicated which part of the Table/guide this statement can be found (see Page 6). Given the dialogic nature of the focus groups and interviews, we felt it important to describe our PrEP explanations in detail, so as to accurately describe our methods.

3. A related point is that the lengthy description of PrEP in the Methods section, taken together with the findings on the "interpreting effectiveness" section of the Results, highlights another key problem that is not discussed enough in the paper – namely, that a key barrier is likely the difficulty that healthcare providers and/or authorities will have in communicating the concept of combination prevention in a clear and understandable way. This concept is briefly alluded to at the beginning of the Discussion ("We identified the significance of how the effectiveness of PrEP... is communicated") but it is mostly absent from the following paragraph, which emphasizes that understanding on the part of consumers is the major "stumbling block". Communication of PrEP efficacy/effectiveness is a two-way process that will require clarity and understanding on the part of both providers and users.

- We fully agree with these comments, and have added in comments to this extent in the second paragraph of the discussion (see page 15 – 16) and in the concluding paragraph of the article (see page 18).

4. A previously raised concern was that not perceiving oneself as a PrEP candidate and low perceptions of HIV risk should not necessarily be labelled as a "barrier" to PrEP, since many people are indeed at low risk of acquiring HIV. The authors have responded to this comment by describing how and why perceptions of HIV risk can be erroneous, assumptive and related to HIV stigma. These points are appropriate and valid, but should be raised directly in the paper; otherwise it still seems incomplete to simply label these issues as "barriers".

- We agree with the reviewers comments in relation to the genuine possibility that some participants were at low risk of acquiring HIV. As we explained in our previous correspondence, a number of participants practiced serosorting based on assumptions about disclosure grounded in HIV stigma.

We have made this point more explicit in the results section (See page 12) in relation to HIV-negative participants descriptions of serosorting, and that some of these strategies seemed to be grounded in HIV-stigma and assumptions about HIV disclosure. We have also added in clarification to the discussion (see page 16) to include consideration of how HIV stigma might affect risk perception. We hope that our change to wording in relation to serosorting or monogamous relationships in the results, and the broad language in the discussion in relation to consideration of risk perception as an important factor in PrEP candidacy, does not explicitly claim that all HIV-negative MSM or Africans should be considered candidates for PrEP. Rather, that these factors should be considered as areas to pay particular attention to when implementing PrEP support.

5. In the concluding paragraph, what is “non-HIV risk reduction”?

- Non-HIV risk reduction refers to the other risks identified by participants which PrEP would not address, such as STIs and pregnancy, which we describe in PrEP and other concerns section of the results on p 12 – 13. We have slightly modified the wording in the paragraph to ‘non-HIV related risk reduction’ to better reflect this. (See page 17)

Reviewer: 3: K R Amico, University of Michigan, USA

Abstract: Consider changing “identified barriers to PrEP uptake and use” to “identified barriers to potential PrEP uptake and use” as the work was focused on intentions and hypothetical use rather than responses to an actual opportunity to use PrEP.

- We have changed the wording as suggested (See Abstract, Results)

Article summary: The authors note that the paper identifies inequalities in HIV literacy but the results of the paper do not suggest inequalities, rather they suggest low levels of accurate PrEP information.

- While we agree that one of the main messages of the paper is not in relation to inequalities in HIV literacy, we felt it was more accurate to specify limited understandings of PrEP effectiveness as a barrier, rather than low levels of information. This is especially the case as participants struggled with interpreting statistical efficacy rates, and understanding how these might work in combination with other risk reduction strategies. We have therefore modified the message accordingly (See page 3).

Introduction: The authors provide a rationale for inclusion of HIV-positive individuals in the focus groups/IDIs but it seems to fall short of reaching a logical conclusion. While serodiscordant partnerships would be a targeted group for PrEP, why attitudes and beliefs from an individual living with HIV are important should rest with their influence in supporting partners or dissuading them. Consider a slight revision to call that out specifically- eg., opinions of HIV-positive individuals will shape PrEP uptake through influence on the community as well as more directly on potential HIV-negative partners. If phrased something like that, there is no need for the sentence that follows (...We did not explicitly...).

- We have changed the wording to reflect the specific reasons for including HIV-positive participants and adopted the suggested wording. (See page 5)

Methods: Consider moving “We have identified extracts...” from method to the results section.

- We have moved this sentence to the results section (See page 7)

Results: Page 11- Consider using a different word than “significant” in “A significant minority...”

- We have used the word large minority here (See page 11).

Results: Page 11- Consider moving “Many participants described skepticism...” to the next section “PrEP candidacy”

- We have moved this sentence to the following section on PrEP candidacy (see page 11).

Results: Page 12- The authors describe discourse here and elsewhere (page 13) as “could not imagine” which has a ring of judgment to it and feels a bit extreme and stigmatizing. Rather, it seems that most adopted serosorting or believed they could both accurately tell if a potential partner was HIV positive and avoid sex with him/her.

- We appreciate this point and have changed the wording accordingly, so that it does not further stigmatize or judge participant risk management strategies (See page 12).

Discussion: In a few places, the authors note that “we demonstrated” but really the study was not designed to demonstrate, rather it was to explore, characterize or identify. Thus, the authors explored, identified, characterized, and results suggest, highlight, identify, so on, but do not conclusively demonstrate.

- We agree with the reviewer and have changed the wording in the discussion to reflect this (See pages 14 – 17).

Discussion: The authors note “Our findings suggest that HIV literacy is directly affected by geography...” It is not clear what findings are being used to support this claim. The results do not provide support for this.

- While this finding emerged in our study, we have not sufficiently demonstrated it in this paper to make this claim. We have therefore removed this element and referenced work which has identified how inequalities in HIV knowledge are affected by proximity to HIV. We have changed the wording to this effect (see page 16).

Discussion- Page 16- typo (line 54)

- We have deleted the unnecessary full stop from the sentence.

Tables: Adding in items from the guides are very helpful. However, presently that table reads like a protocol, or procedural document that the interviewer/facilitator would have used for implementation. The reader does not need that level of detail. Readers will want to be able to see either a list of questions or a select example of the kinds of items asked. This should not take up several pages but rather be an actual table with themes and sample items. The authors should consolidate current text to make this a brief set of questions or set of example questions for major areas of inquiry.

- We have modified the tables, so as to present the main topic areas and questions in each of the discussion guides. We have kept our explanation of PrEP and TasP within the Tables as we felt it important to give an indication of the nature of our explanation to participants.

We would be grateful if you would again consider this paper for publication and look forward to receiving your reply.